# Direct Electrooxidative Selenylation/Cyclization of Alkynes: Access to Functionalized Benzo[*b*]furans

**DOI:** 10.3390/molecules27196314

**Published:** 2022-09-25

**Authors:** Balati Hasimujiang, Shengsheng Lin, Chengwei Zheng, Yong Zeng, Zhixiong Ruan

**Affiliations:** Guangzhou Municipal and Guangdong Provincial Key Laboratory of Molecular Target & Clinical Pharmacology, the NMPA and State Key Laboratory of Respiratory Disease, School of Pharmaceutical Sciences and the Fifth Affiliated Hospital, Guangzhou Medical University, Guangzhou 511436, China

**Keywords:** eletrochemical synthesis, organoselenium compounds, benzo[*b*]furans, cyclization, electrooxidation

## Abstract

A mild, practical, metal and oxidant-free methodology for the synthesis of various C-3 selenylated benzo[*b*]furan derivatives was developed through the intramolecular cyclization of alkynes promoted with diselenides via electrooxidation. A wide range of selenium-substituted benzo[*b*]furan derivatives were obtained in good to excellent yields with high regioselectivity under constant current in an undivided cell equipped with carbon and platinum plates as the anode and cathode, respectively. Moreover, the convergent approach exhibited good functional group tolerance and could be easily scaled up with good efficiency, providing rapid access to a diverse range of selenylated benzo[*b*]furans derivatives from simple, readily available starting materials.

## 1. Introduction

Organoselenium compounds, especially heterocycle motifs containing selenium, frequently play an important role in pharmaceutical, biological, and material applications as well as in modern organic synthesis because of its specific chemical and biological properties [1,2,3,4,5,6]. Selenium is an indispensable trace element in the human body; the introducing of the Se atom into organic molecules can modify their chemical and physical properties. Organoselenium compounds play an important role not only in organic transformation but also in pharmaceutical, biological, and material applications, especially selenium compounds containing heterocyclic units have unique biological activities [7,8,9], such as antitumor, antibacterial, anti-inflammatory, antiviral, cardiovascular protection, immune regulation, and other biological activities (Figure 1a). In particular, the selenylation of heteroarenes including the benzofuran moiety is a very important class of heterocycles widely found in natural products, dyes, materials, and pharmaceutical ingredients. Among the numerous benzofuran derivatives, 3-substituted benzo[*b*]furans display a wide array of biological activities that include anti-inflammatory [10,11], antibacterial [12,13], antimicrobial [14], antitumor [15,16], anticonvulsant [17,18], etc. Some representative examples of bioactive benzo[*b*]furans are outlined in Figure 1 (Figure 1b). Therefore, the development of an efficient and general method for the synthesis of selenated benzo[*b*]furans is still highly desired.

Over the past few decades, some significant progress has been made in the field of synthesis of organoselenium compounds [19,20,21,22,23,24,25] as well as benzofurans. Most of the transformations were traditionally utilized transition-metal catalysts or stoichiometric amounts of oxidants. For example, pioneering work toward the synthesis of selenated benzofurans relied on an electrophilic cyclization strategy using toxic and instable RSeCl as the selenium source [26,27]. Most of the research to access selenylbenzofurans mainly focused on the transition metals, such as the requirement of precious palladium catalyst or stoichiometric amount of Fe/Cu/Ag catalysts (Figure 1a) [28,29,30,31,32,33]. Recently, Frank and co-workers reported a catalytic strategy of direct C–H thiolation of heteroarenes to obtain selenylbenzofurans by a heterogeneous catalyst Pd/Al_2_O_3_, but it suffered from low yields and poor selectivity, particularly for the formation of selenylbenzofurans (Figure 1b) [34]. Despite the significant achievements that have been made in this field, there are still some limitations, such as these methods involving transition metals, excess external oxidants, low reaction yields and poor selectivity. Undoubtedly, the development of an efficient, practical and greener procedure for the formation of selenylated benzofuran building blocks involving simple operation under mild reaction conditions is still urgent and highly desirable.

In recent years, organic electrochemical synthesis has become an attractive approach in the field of organic synthesis because of its environmentally benign, sustainable, and practical nature [35,36,37,38,39,40,41,42]. Recently, several electrochemically induced radical functionalizations of alkynes have been reported [43,44,45,46]. Inspired by the recent developments in the area of electrochemical selenylation and our continuous research interest in the electrochemical synthesis [47,48,49,50,51], herein, we developed an electrochemical selenylation/cyclization method of alkynes for the synthesis of 3-selenylated benzofuran motifs with wide functional groups under mild electrochemical anodic oxidation conditions in an undivided cell (Figure 1c).

## 2. Results and Dissections

### 2.1. Optimization of Reaction Conditions

We started our study by employing 2-alkynylanisole **1a** and diphenyl diselenide (**2a**) as model substrates to optimize the reaction conditions and explore the key variables (Table 1). When using the co-solvent of acetonitrile and hexafluoroisopropanol (HFIP), the desired product **3a** could be delivered in an excellent yield of 96% in an undivided cell equipped with a graphite anode and a platinum plate cathode under a constant current of 10 mA in 2 h without any other additives in the presence of supporting electrolytes *n*-Bu_4_NPF_6_ at room temperature (Table 1, entry 1). The use of MeOH instead of the co-solvent of MeCN and HFIP delivered a low yield of desired product (entry 2). Upon use of MeCN (entry 3) as a sole solvent, a slightly decreased yield was obtained. To identify the electrolyte potentially suitable for the cyclization, various electrolytes were tested. The use of *n*-Bu_4_NClO_4_, *n*-Bu_4_NBF_4_, and LiClO_4_ instead of *n*-Bu_4_NPF_6_ delivered lower yields (entries 4–6), while the use of *n*-Bu_4_NPF_6_ gave the best yield of the desired product (entry 1). Decreasing current to 5 mA afforded an obviously decreased yield of 87% in a longer time (entry 7). In the end, we explored the electrode materials for this electrochemical cyclization reaction. C(+)/C(−) electrodes led to diminished yields, while the Pt(+)/Pt(−) combination gave a very similar yield (entries 8 and 9). The use of 0.6 equivalent of diphenyl diselenide (**2a**) proved to be inefficient, delivering product **3a** with 81% yield (entry 10). No desired product was observed without the electrolyte or electricity (entries 11 and 12).

### 2.2. Substrate Scope

With the optimal reaction conditions in hand, we began to investigate the generality of this electrochemical protocol. A range of 2-alkynylanisoles bearing various substituents were explored under the standard reaction conditions, and the scope was summarized in Figure 2. It was found that the R group bearing both an electron-donating group such as alkyl groups and an electron-withdrawing group such as halogens on the benzene ring was amenable to the mild electrochemical selenylation/cyclization and afforded the desired products in good to excellent yields 85−96% (**3a**–**3j**). When the R was installed with a fused ring, such as naphthalene or even anthracene, the corresponding selenylated products **3k** and **3l** were still obtained in 82% and 94% yields, respectively. The structure of **3l** was unambiguously confirmed by single crystal X-ray diffraction studies. While R was installed with aromatic heterocycles, pyridine and the thiophene ring were also suitable for the reaction, which gave the desired products **3m** in 45% and **3o** in 83% yields, respectively. It is particularly noteworthy that the electrochemical cyclization smoothly delivered the desired selenylated products in excellent yields when R was replaced by the hydroxyl alkyl groups (**3p**). Subsequently, the scope of the aryl ring with various substituents was tested. It was found that halogen and alkyl groups on the aryl ring as well as heterocycles were also proved to be tolerated under the standard reaction conditions and delivered the desired functionalized selenylbenzofurans in good to excellent yields (**3****q**–**3****u**). Notably, the 2-[(trimethylsilyl)ethynyl]anisole (**1v**) proved to be a suitable substrate, which gave the bis-selenylated benzofuran product **3v** in moderate yield.

We next continued to speculate the cyclization of 2-alkynylanisole **1a** with different diorganyl diselenides **2** (Figure 3). The selenylation/cyclization regime proved to be tolerant to electron-donating and electron-withdrawing groups on the aromatic rings bonded to the selenium atom of the diaryl diselenides **2**, giving the corresponding 3-organoselanyl-benzo[*b*]furans **4** in good yields (**4a**–**4j**). Gratifyingly, the electrooxidative cyclization approach showed toleration to the presence of heterocyclic diselenide, furnishing the benzo[*b*]furan derivative **4k** in a good yield of 83%. It is worth mentioning that the reaction demonstrated satisfactory tolerance not only with aryl diselenides but also alphatic diselenide derivatives to prepare highly functionalized benzo[*b*]furans. For example, dimethyl diselenide (**2l**) and diethyl diselenide (**2m**) smoothly produced the selenylated products **4l** and **4m** in good yields.

### 2.3. Scaled-Up Synthesis and Transformation

To prove the synthetic value of this electrochemical selenylation/cyclization reaction, scaled-up reactions of **1a** and **1v** were carried out with **2a** under standard conditions. As shown in Figure 4, the scale-up reactions afforded the corresponding compounds **3a** (4.5 g) and **3****t** (1.85 g) in a high reaction efficiency in slightly reduced yields (85% and 84%) without an appreciable loss in efficacy.

### 2.4. Control Experiments and Plausible Mechanism

For a deeper understanding of the mechanism of this reaction, some control experiments were designed (Figure 5). The solo product **3w** of selenylation/cyclization with a phenol group was obtained by the intramolecular competition experiment, indicating the reaction preferred the single electron transfer (SET) (Figure 5a, ***Pathway I***). When electrolysis was performed in the presence of the radical scavenger 2,6-di-tert-butyl-4-methylphenol (BHT) under the standard reaction conditions, the desired product **3a** had an obviously decreased yield of 57% (Figure 5b), which implied that the process was likely proceeded by a favored radical pathway (***Pathway I***). Subsequently, when substrate **1a** was treated with PhSeCl instead of diphenyldiselenide in MeCN/HFIP at room temperature, the desired product **3a** was obtained in 48% yield (Figure 5c) in 2 h, without electric current, which indicates that the electrochemical selenylation/cyclization process could not exclude the ionic pathway (***Pathway II***). Finally, we investigated the effect of the substituents on the oxygen atom, and it showed that when the oxygen atom related H instead of the methyl group, the reaction was shown to have good tolerance with an 86% yield of **3a**, whereas a phenyl group failed to deliver the desired product **3****x** with a recovery of **1x** in 96% (Figure 5d). The reaction mixture was monitored by GC-MS analysis (Appendix A), and the molecular weight of intermediate PhSeMe was captured. Thus, this reaction was initiated by the formation of seleno radical **B** and phenyl selenium anion **H** through cathode reduction. Then, the selenium anion **H** was reacted with methyl cation to produce PhSeMe (Figure 6). These control experiments are suggesting that the diselenide can be involved in both processes (anodic oxidation and cathode reduction), producing seleno radical **B** and seleno cation **C** under electrochemical conditions.

Furthermore, cyclic voltammetry (CV) experiments were also performed to reveal key mechanistic insights into the electrochemical transformation (Figure 2). An obvious oxidation peak of substrate **1a** could be observed at 1.610 V, which was higher than that of diphenyl diselenide (**2a**, 1.388 V), demonstrating that **2a** should be preferentially oxidized at the anode in the electrochemical system.

Thus, based on the enriched evidence from the control experiments and previous related reports [52,53,54], a possible mechanism for the electrochemcial selenylation/cyclization is described in Figure 6. At the beginning, the reaction was initiated by the formation of seleno radical **B** and seleno cation **C** via a cathode reduction and the anodic oxidation of diphenyl diselenide (**2a**). Thereafter, the radical addition of **B** to **1a** generates the radical intermediate **D**, intermediate **D** undergoes cyclization to give the intermediate **E**, intermediate **E** followed by electron loss afforded intermediate **F**. Alternatively, the ionic pathway cannot be omitted; that is, the selenium cation **C** can also add to **1a** to provide the cationic intermediate **G**, and intermediate **G** undergoes cyclization to give the intermediate **F**, which then undergoes demethylation to produce the desired product **3a** and PhSeMe with selenium anions.

## 3. Conclusions

In summary, we have successfully developed an efficient electrochemical protocol for the construction of valuable selenylated benzo[*b*]furan derivatives through an electrooxidative process by the cyclization of 2-alkynylanisoles. Various 3-substituted benzofurans were obtained in good to excellent yields under metal- and oxidant-free mild reaction conditions. Furthermore, the current protocol was successfully applied to gram scale, and a conceivable reaction mechanism was proposed.

## 4. Materials and Methods

### 4.1. General Information

Electrochemical reactions were conducted using an AXIOMET AX-3003P potentiostat in constant current mode using an undivided cell equipped with a graphite plate (1.5 cm × 1.0 cm × 0.2 cm) as the anode and platinum plate (1.0 cm × 1.0 cm × 0.01 cm) as the cathode under air. Graphite plates are commercially available from Beijing Jinglong Special Carbon Technology Co., Ltd (Beijing, China). Platinum electrodes are commercially available from Tianjin Aida (Tianjin, China). The substrate **1a**–**1****x** and diselenides were synthesized according to previously described methods. Other chemicals were obtained from commercial sources and were used without further purification. Yields refer to isolated compounds, which were estimated to be >95% pure as determined by ^1^H-NMR. TLC: Macherey-Nagel, TLC plates Alugram^®^Sil G/UV254. Detection under UV light at 254 nm. Chromatography separations were carried out on silica gel 60H (200–300 mesh) manufactured by Qingdao Haiyang Chemical Group Co. (Qingdao, China). High-resolution mass spectrometry (HRMS) was measured on Thermo-DFS mass spectrometer. NMR spectra were recorded on JEOL 400 NMR (^1^H 400 MHz; ^13^C 100 MHz; ^19^F 376 MHz) in CDCl_3_. If not otherwise specified, chemical shifts (*δ*) are given in ppm.

### 4.2. Materials

The starting materials **1a**–**1****x** were prepared according to the reported methods [54]. All the solvents are commercially available and directly used in this electrochemical synthesis.

### 4.3. Procedure for the Electrosynthesis of Compounds ***3***

In an undivided cell (10 mL) equipped with a stirring bar, a mixture of substrates **1** (0.3 mmol), **2a** (0.36 mmol), *n*-Bu_4_NPF_6_ (0.5 mmol) and MeCN/HFIP = 4:1 (5 mL) was added. The cell was equipped with a graphite plate (1.5 cm × 1.0 cm × 0.2 cm) as the anode and platinum plate (1.0 cm × 1.0 cm × 0.01 cm) as the cathode connected to an AXIOMET AX-3003P DC regulated power supply. The reaction mixture was stirred and electrolyzed at a constant current of 10 mA at room temperature for 2 h. Upon completion, the solvent was removed directly under reduced pressure to afford the crude product, which was further purified by flash column chromatography to afford the desired products **3a**–**3****w**.

*2-Phenyl-3-(phenylselanyl)benzofuran* (**3a**): Prepared following general procedure **A** using substrate 1-methoxy-2-(phenylethynyl)benzene **1a** (0.3 mmol, 62 mg) and diphenyl diselenide **2a** (0.36 mmol, 112 mg). Isolation was purified by flash chromatography and eluted with petroleum ether to give a yellow solid **3a** (101 mg, 96%). Mp: 62.0–63.8 °C. ^1^H NMR (400 MHz, CDCl_3_) *δ* = 8.22 (d, *J* = 7.5 Hz, 2H), 7.55 (dd, *J* = 16.3, 8.0 Hz, 2H), 7.46 (m, 2H), 7.43–7.34 (m, 2H), 7.33–7.28 (m, 2H), 7.25–7.21 (m, 1H), 7.16 (m, 3H). ^13^C NMR (100 MHz, CDCl_3_) *δ* = 157.4, 154.3, 132.1, 131.5, 130.3, 129.4, 129.4, 129.3, 128.6, 127.9, 126.4, 125.4, 123.6, 121.4, 111.3, 99.8. Analytical data for compound **3a** were consistent with the literature [55].

*3-(Phenylselanyl)-2-(p-tolyl)benzofuran* (**3b**): Prepared following general procedure **A** using substrate 1-methoxy-2-(*p*-tolylethynyl)benzene **1b** (0.3 mmol, 67 mg) and diphenyl diselenide **2a** (0.36 mmol, 112 mg). Isolation was purified by flash chromatography and eluted with petroleum ether to give a yellow solid **3b** (104 mg, 95%). Mp: 84.1–85.3 °C. ^1^H NMR (400 MHz, CDCl_3_) *δ* = 8.14–8.10 (m, 2H), 7.54 (dd, *J* = 16.3, 8.0 Hz, 2H), 7.39–7.25 (m, 4H), 7.29–7.18 (m, 2H), 7.23–7.08 (m, 3H), 2.41 (s, 3H). ^13^C NMR (100 MHz, CDCl_3_) *δ* = 157.7, 154.1, 139.6, 132.1, 131.7, 129.4, 129.3, 129.2, 127.8, 127.4, 126.3, 125.1, 123.5, 121.2, 111.2, 99.0, 21.6. Analytical data for compound **3b** were consistent with the literature [55].

*2-(4-Ethylphenyl)-3-(phenylselanyl)benzofuran* (**3c**): Prepared following general procedure A using substrate 1-((4-ethylphenyl)ethynyl)-2-methoxybenzene **1c** (0.3 mmol, 71 mg) and diphenyl diselenide **2a** (0.36 mmol, 112 mg). Isolation was purified by flash chromatography and eluted with petroleum ether to give a yellow solid **3c** (106 mg, 94%). Mp: 66.3–67.5 °C. ^1^H NMR (400 MHz, CDCl_3_) *δ* = 8.17–8.13 (m, 2H), 7.58–7.51 (m, 2H), 7.36–7.28 (m, 5H), 7.24 (m, 1H), 7.20–7.12 (m, 3H), 2.71 (q, *J* = 7.5 Hz, 2H), 1.28 (t, *J* = 7.6 Hz, 3H). ^13^C NMR (100 MHz, CDCl_3_) *δ* = 157.8, 154.2, 145.9, 132.1, 131.7, 129.4, 129.2, 128.2, 127.9, 127.7, 126.3, 125.1, 123.5, 121.2, 111.2, 99.0, 28.9, 15.5. ^77^Se NMR (115 MHz, CDCl_3_) *δ* = 462.5. HR-MS (ESI) *m*/*z* calcd for C_22_H_19_OSe [M + H]^+^ 379.0596, found 379.0521.

*2-(4-Fluorophenyl)-3-(phenylselanyl)benzofuran* (**3d**): Prepared following general procedure **A** using substrate 1-((4-fluorophenyl)ethynyl)-2-methoxybenzene **1d** (0.3 mmol, 68 mg) and diphenyl diselenide **2a** (0.36 mmol, 112 mg). Isolation was purified by flash chromatography and eluted with petroleum ether to give a yellow solid **3d** (100 mg, 91%). Mp: 78.8–79.5 °C. ^1^H NMR (400 MHz, CDCl_3_) *δ* = 8.22 (m, 2H), 7.55 (m, 2H), 7.35 (m, 1H), 7.31–7.27 (m, 2H), 7.25 (m, 1H), 7.21–7.12 (m, 5H). ^13^C NMR (100 MHz, CDCl_3_) *δ* = 163.4 (d, ^1^*J*_C-F_ = 249.0 Hz), 155.4 (d, ^1^*J*_C-F_ = 234.0 Hz), 132.0, 131.4, 130.0, 129.9, 129.5, 129.3, 129.2, 126.5, 125.4, 123.7, 121.3, 115.7 (d, ^2^*J*_C-F_ = 22.0 Hz), 111.3, 99.5. ^19^F NMR (376 MHz, CDCl_3_) *δ* = −110.7. Analytical data for compound **3d** were consistent with the literature [32].

*2-(4-Chlorophenyl)-3-(phenylselanyl)benzofuran* (**3e**): Prepared following general procedure **A** using substrate 1-((4-chlorophenyl)ethynyl)-2-methoxybenzene **1e** (0.3 mmol, 73 mg) and diphenyl diselenide **2a** (0.36 mmol, 112 mg). Isolation was purified by flash chromatography and eluted with petroleum ether to give a yellow solid **3e** (107 mg, 93%). Mp: 77.3–79.0 °C. ^1^H NMR (400 MHz, CDCl_3_) *δ* = 8.14 (d, *J* = 8.4 Hz, 2H), 7.51 (m, 2H), 7.39 (d, *J* = 8.5 Hz, 2H), 7.32 (m, 1H), 7.26–7.19 (m, 3H), 7.13 (m, 3H). ^13^C NMR (100 MHz, CDCl_3_) *δ* = 156.1, 154.2, 135.4, 132.0, 131.2, 129.5, 129.3, 129.1, 128.9, 128.7, 126.6, 125.6, 123.7, 121.4, 111.3, 100.4. Analytical data for compound **3e** were consistent with the literature [32].

*2-(4-Bromophenyl)-3-(phenylselanyl)benzofuran* (**3f**): Prepared following general procedure **A** using substrate 1-((4-bromophenyl)ethynyl)-2-methoxybenzene **1f** (0.3 mmol, 86 mg) and diphenyl diselenide **2a** (0.36 mmol, 112 mg). Isolation was purified by flash chromatography and eluted with petroleum ether to give a yellow solid **3f** (116 mg, 90%). Mp: 91.2–91.5 °C. ^1^H NMR (400 MHz, CDCl_3_) *δ* = 8.08 (d, *J* = 8.6 Hz, 2H)., 7.58–7.45 (m, 4H), 7.32 (m, 1H), 7.26–7.18 (m, 3H), 7.17–7.08 (m, 3H). ^13^C NMR (100 MHz, CDCl_3_) *δ* = 156.1, 154.2, 132.0, 131.8, 131.2, 129.5, 129.4, 129.3, 129.3, 129.1, 126.6, 125.7, 123.7, 121.4, 111.3, 100.5. Analytical data for compound **3f** were consistent with the literature [32].

*N-(4-(3-(Phenylselanyl)benzofuran-2-yl)phenyl)acetamide* (**3g**): Prepared following general procedure **A** using substrate *N*-(4-((2-methoxyphenyl)ethynyl)phenyl)- acetamide **1g** (0.3 mmol, 80 mg) and diphenyl diselenide **2a** (0.36 mmol, 112 mg). Isolation was purified by flash chromatography and eluted with petroleum ether to give a yellow oil **3g** (110 mg, 90%). ^1^H NMR (400 MHz, DMSO-*d*_6_) *δ* = 10.15 (s, 1H), 8.08–7.96 (m, 2H), 7.68 (d, *J* = 8.7 Hz, 2H), 7.62 (d, *J* = 8.1 Hz, 1H), 7.39–7.25 (m, 2H), 7.23–7.18 (m, 3H), 7.17–7.07 (m, 3H), 2.02 (s, 3H). ^13^C NMR (100 MHz, DMSO-*d*_6_) *δ* = 168.7, 156.8, 153.4, 140.7, 131.3, 130.8, 129.6, 128.9, 128.1, 126.6, 125.4, 123.8, 123.7, 120.5, 118.8, 111.4, 98.2, 24.2. HR-MS (ESI) *m*/*z* calcd for C_22_H_17_NNaO_2_Se [M + Na]^+^ 430.0317, found 430.0315.

*2-(3-Fluorophenyl)-3-(phenylselanyl)benzofuran* (**3h**): Prepared following general procedure **A** using substrate 1-((3-fluorophenyl)ethynyl)-2-methoxybenzene **1h** (0.3 mmol, 68 mg) and diphenyl diselenide **2a** (0.36 mmol, 112 mg). Isolation was purified by flash chromatography and eluted with petroleum ether to give a yellow solid **3h** (94 mg, 85%). Mp: 59.6–60.8 °C. ^1^H NMR (400 MHz, CDCl_3_) *δ* = 8.04 (d, *J* = 7.9 Hz, 1H), 7.97 (d, *J* = 10.5 Hz, 1H), 7.54 (dd, *J* = 10.6, 8.1 Hz, 2H), 7.44–7.32 (m, 2H), 7.31–7.27 (m, 2H), 7.23 (d, *J* = 7.1 Hz, 1H), 7.16 (m, *J* = 3.3 Hz, 3H), 7.08 (m, 1H). ^13^C NMR (100 MHz, CDCl_3_) *δ* = 162.8 (d, ^1^*J*_C-F_ = 244.0 Hz), 154.9 (d, ^1^*J*_C-F_ = 145.0 Hz), 132.2 (d, ^3^*J*_C-F_ = 9.0 Hz), 131.9, 131.1, 130.2 (d, ^3^*J*_C-F_ = 9.0 Hz), 129.5, 129.5, 129.3, 126.6, 125.8, 123.7, 123.5 (d, ^4^*J*_C-F_ = 2.0 Hz), 121.5, 116.3 (d, ^2^*J*_C-F_ = 21.0 Hz), 114.7 (d, ^2^*J*_C-F_ = 24.0 Hz), 111.4, 101.1. ^19^F NMR (376 MHz, CDCl_3_) *δ* = −112.1. ^77^Se NMR (115 MHz, CDCl_3_) *δ* = 462.4. HR-MS (ESI) *m*/*z* calcd for C_20_H_14_FOSe [M + H]^+^ 369.0188, found 369.0112.

*2-(2-Chlorophenyl)-3-(phenylselanyl)benzofuran* (**3i**): Prepared following general procedure **A** using substrate 1-chloro-2-((2-methoxyphenyl)ethynyl)benzene **1i** (0.3 mmol, 73 mg) and diphenyl diselenide **2a** (0.36 mmol, 112 mg). Isolation was purified by flash chromatography and eluted with petroleum ether to give a slight yellow solid **3i** (102 mg, 89%). Mp: 84.0–85.0 °C. ^1^H NMR (400 MHz, CDCl_3_) *δ* = 7.60 (d, *J* = 8.2 Hz, 1H), 7.57–7.49 (m, 3H), 7.46–7.32 (m, 3H), 7.32–7.27 (m, 3H), 7.21–7.13 (m, 3H). ^13^C NMR (100 MHz, CDCl_3_) *δ* = 156.8, 155.0, 134.7, 132.9, 131.1, 130.4, 130.2, 129.8, 129.4, 129.3, 126.5, 126.4, 125.5, 123.6, 121.4, 111.7, 104.0. Analytical data for compound **3i** were consistent with the literature [32].

*2-(2-Bromophenyl)-3-(phenylselanyl)benzofuran* (**3j**): Prepared following general procedure **A** using substrate 1-bromo-2-((2-methoxyphenyl)ethynyl)benzene **1j** (0.3 mmol, 86 mg) and diphenyl diselenide **2a** (0.36 mmol, 112 mg). Isolation was purified by flash chromatography and eluted with petroleum ether to give a yellow oil **3j** (116 mg, 90%). ^1^H NMR (400 MHz, CDCl_3_) *δ* = 7.72 (d, *J* = 7.9 Hz, 1H), 7.60 (d, *J* = 8.2 Hz, 1H), 7.51 (d, *J* = 7.6 Hz, 2H), 7.42–7.33 (m, 3H), 7.29 (m, 3H), 7.20–7.14 (m, 3H). ^13^C NMR (100 MHz, CDCl_3_) *δ* = 158.0, 154.8, 133.3, 133.0, 131.5, 131.3, 131.1, 130.3, 129.9, 129.2, 127.1, 126.4, 125.4, 124.3, 123.6, 121.4, 111.7, 103.8. HR-MS (ESI) *m*/*z* calcd for C_20_H_14_BrOSe [M + H]^+^ 428.9388, found 428.9383.

*2-(Naphthalen-2-yl)-3-(phenylselanyl)benzofuran* (**3k**): Prepared following general procedure **A** using substrate 2-((2-methoxyphenyl)ethynyl)naphthalene **1k** (0.3 mmol, 77 mg) and diphenyl diselenide **2a** (0.36 mmol, 112 mg). Isolation was purified by flash chromatography and eluted with petroleum ether to give a yellow solid **3k** (98 mg, 82%). Mp: 121.7–123.4 °C. ^1^H NMR (400 MHz, CDCl_3_) *δ* = 8.71 (s, 1H), 8.40 (d, *J* = 8.5 Hz, 1H), 7.91 (d, *J* = 8.3 Hz, 2H), 7.89–7.83 (m, 1H), 7.60 (dd, *J* = 16.0, 8.0 Hz, 2H), 7.53 (m, 2H), 7.37 (m, 3H), 7.32–7.24 (d, *J* = 7.5 Hz, 1H), 7.18 (m, 3H). ^13^C NMR (100 MHz, CDCl_3_) *δ* = 157.3, 154.4, 133.6, 133.2, 132.2, 131.6, 129.5, 129.5, 128.9, 128.2, 127.8, 127.6, 127.1, 126.6, 126.5, 125.4, 124.9, 123.6, 121.4, 111.3, 100.4. Analytical data for compound **3k** were consistent with the literature [32].

*2-(Phenanthren-9-yl)-3-(phenylselanyl)benzofuran* (**3l**): Prepared following general procedure **A** using substrate 9-((2-methoxyphenyl)ethynyl)phenanthrene **1l** (0.3 mmol, 93 mg) and diphenyl diselenide **2a** (0.36 mmol, 112 mg). Isolation was purified by flash chromatography and eluted with petroleum ether to give a yellow solid **3l** (127 mg, 94%). Mp: 146.5–148.2 °C. ^1^H NMR (400 MHz, CDCl_3_) *δ* = 8.76 (dd, *J* = 17.8, 8.3 Hz, 2H), 8.03 (d, *J* = 8.3 Hz, 1H), 7.94 (s, 1H), 7.85 (d, *J* = 7.8 Hz, 1H), 7.66 (m, 6H), 7.44 (m, 1H), 7.38–7.31 (m, 3H), 7.20–7.11 (m, 3H). ^13^C NMR (100 MHz, CDCl_3_) *δ* = 159.1, 155.0, 131.8, 131.6, 131.2, 131.0, 130.8, 130.8, 130.7, 130.2, 129.5, 129.3, 128.0, 127.1, 127.0, 126.8, 126.5, 126.1, 125.4, 123.7, 123.1, 122.8, 121.5, 111.7, 104.5. The compound **3l** cannot be ionized in ESI and APCI.

*2-(3-(Phenylselanyl)benzofuran-2-yl)pyridine* (**3m**): Prepared following general procedure **A** using substrate 2-((2-methoxyphenyl)ethynyl)pyridine **1m** (0.3 mmol, 63 mg) and diphenyl diselenide **2a** (0.36 mmol, 112 mg). Isolation was purified by flash chromatography (PE/EtOAc: 30/1→10/1) to give a yellow oil **3m** (47 mg, 45%). ^1^H NMR (400 MHz, CDCl_3_) *δ* = 8.14 (d, *J* = 8.6 Hz, 2H), 7.58 (dd, *J* = 17.4, 8.1 Hz, 4H), 7.38 (m, 1H), 7.34–7.27 (m, 3H), 7.20 (m, 3H). ^13^C NMR (100 MHz, CDCl_3_) *δ* = 156.1, 154.2, 131.9, 131.8, 131.2, 129.5, 129.4, 129.3, 129.2, 129.1, 126.5, 125.6, 123.7, 121.4, 111.3, 100.5. HR-MS (ESI) *m*/*z* calcd for C_19_H_14_NOSe [M + H]^+^ 352.0235, found 352.0234.

*3-(Phenylselanyl)-2-(thiophen-2-yl)benzofuran* (**3n**): Prepared following general procedure **A** using substrate 2-((2-methoxyphenyl)ethynyl)thiophene **1n** (0.3 mmol, 64 mg) and diphenyl diselenide **2a** (0.36 mmol, 112 mg). Isolation was purified by flash chromatography and eluted with petroleum ether to give a yellow solid **3n** (86 mg, 81%). Mp: 84.1–85.4 °C. ^1^H NMR (400 MHz, CDCl_3_) *δ* = 7.95–7.86 (m, 1H), 7.52 (m, 2H), 7.45–7.37 (m, 1H), 7.32 (m, 3H), 7.28–7.19 (m, 1H), 7.22–7.08 (m, 4H). ^13^C NMR (100 MHz, CDCl_3_) *δ* = 154.0, 131.9, 131.9, 131.2, 129.6, 129.5, 129.4, 128.1, 127.8, 127.5, 126.5, 125.3, 123.7, 120.9, 111.1, 99.2. HR-MS (ESI) *m*/*z* calcd for C_18_H_13_OSSe [M + H]^+^ 356.9847, found 356.9843.

*3-(Phenylselanyl)-2-(thiophen-3-yl)benzofuran* (**3o**): Prepared following general procedure **A** using substrate 3-((2-methoxyphenyl)ethynyl)thiophene **1o** (0.3 mmol, 64 mg) and diphenyl diselenide **2a** (0.36 mmol, 112 mg). Isolation was purified by flash chromatography and eluted with petroleum ether to give a yellow solid **3o** (88 mg, 83%). Mp: 114.8–115.5 °C. ^1^H NMR (400 MHz, CDCl_3_) *δ* = 8.22–8.17 (m, 1H), 8.02–7.95 (m, 1H), 7.59–7.48 (m, 2H), 7.38 (dd, *J* = 5.1, 3.0 Hz, 1H), 7.35–7.28 (m, 3H), 7.23 (d, *J* = 7.6 Hz, 1H), 7.20–7.13 (m, 3H). ^13^C NMR (100 MHz, CDCl_3_) *δ* = 154.7, 154.0, 131.9, 131.4, 131.4, 129.5, 129.3, 129.2, 126.8, 126.4, 126.0, 125.4, 125.2, 123.6, 121.1, 111.2. HR-MS (ESI) *m*/*z* calcd for C_18_H_13_OSSe [M + H]^+^ 356.9847, found 356.9843.

*5-(3-(Phenylselanyl)benzofuran-2-yl)pentan-1-ol* (**3p**): Prepared following general procedure **A** using substrate 7-(2-methoxyphenyl)hept-6-yn-1-ol **1p** (0.3 mmol, 65 mg) and diphenyl diselenide **2a** (0.36 mmol, 112 mg). Isolation was purified by flash chromatography and eluted with petroleum ether to give a yellow oil **3p** (96 mg, 89%). ^1^H NMR (400 MHz, CDCl_3_) *δ* = 7.46 (m, 1H), 7.29 (d, *J* = 7.4 Hz, 1H), 7.23 (m, 3H), 7.17 (m, 3H), 3.57 (t, *J* = 6.5 Hz, 2H), 3.01 (t, *J* = 7.5 Hz, 2H), 1.77 (m, 2H), 1.57 (m, 2H), 1.38 (m, 2H), 1.27 (s, 1H). ^13^C NMR (100 MHz, CDCl_3_) *δ* = 163.5, 154.6, 132.0, 130.8, 129.3, 129.2, 126.2, 124.3, 123.3, 120.5, 111.1, 100.4, 62.9, 32.5, 28.1, 27.4, 25.3. HR-MS (ESI) *m*/*z* calcd for C_19_H_21_O_2_Se [M + H]^+^ 361.0701, found 361.0729.

*4-Fluoro-2-phenyl-3-(phenylselanyl)benzofuran* (**3q**): Prepared following general procedure **A** using substrate 1-fluoro-3-methoxy-2-(phenylethynyl)benzene **1q** (0.3 mmol, 68 mg) and diphenyl diselenide **2a** (0.36 mmol, 112 mg). Isolation was purified by flash chromatography and eluted with petroleum ether to give a yellow solid **3q** (98 mg, 89%). Mp: 115.8–116.5 °C. ^1^H NMR (400 MHz, CDCl_3_) *δ* = 8.16 (d, *J* = 7.3 Hz, 2H), 7.49–7.37 (m, 3H), 7.35 (m, 3H), 7.25–7.21 (m, 1H), 7.17 (m, 3H), 6.93–6.84 (m, 1H). ^13^C NMR (100 MHz, CDCl_3_) *δ* = 157.8 (d, ^2^*J*_C-F_ = 23.0 Hz), 156.0 (d, ^2^*J*_C-F_ = 59.0 Hz), 155.4, 132.4, 129.7, 129.5, 129.5, 129.4, 128.5, 128.2, 126.5, 125.5 (d, ^3^*J*_C-F_ = 8.0 Hz), 120.4 (d, ^2^*J*_C-F_ = 16.0 Hz), 109.8 (d, ^2^*J*_C-F_ = 19.0 Hz), 107.6 (d, ^4^*J*_C-F_ = 4.0 Hz), 96.5. ^19^F NMR (376 MHz, CDCl_3_) *δ* = −123.1. HR-MS (ESI) *m*/*z* calcd for C_20_H_14_FOSe [M + H]^+^ 369.0188, found 368.0112.

*5-Methyl-2-phenyl-3-(phenylselanyl)benzofuran* (**3r**): Prepared following general procedure **A** using substrate 1-methoxy-4-methyl-2-(phenylethynyl)benzene **1r** (0.3 mmol, 67 mg) and diphenyl diselenide **2a** (0.36 mmol, 112 mg). Isolation was purified by flash chromatography and eluted with petroleum ether to give a yellow solid **3r** (105 mg, 96%). Mp: 105.2–106.5 °C. ^1^H NMR (400 MHz, CDCl_3_) *δ* = 8.23 (d, *J* = 7.7 Hz, 2H), 7.47 (m, 3H), 7.40 (m, 1H), 7.35 (s, 1H), 7.32 (m, 2H), 7.19 (m, 4H), 2.44 (s, 3H). ^13^C NMR (100 MHz, CDCl_3_) *δ* = 157.6, 152.7, 133.2, 132.1, 131.8, 130.4, 129.5, 129.3, 129.0, 128.6, 127.9, 126.7, 126.2, 121.0, 110.8, 99.3, 21.5. ^77^Se NMR (115 MHz, CDCl_3_) *δ* = 462.5. HR-MS (ESI) *m*/*z* calcd for C_21_H_17_O_2_Se [M + H]^+^ 365.0439, found 365.0438.

*5-Chloro-2-phenyl-3-(phenylselanyl)benzofuran* (**3s**): Prepared following general procedure **A** using substrate 4-chloro-1-methoxy-2-(phenylethynyl)benzene **1s** (0.3 mmol, 73 mg) and diphenyl diselenide **2a** (0.36 mmol, 112 mg). Isolation was purified by flash chromatography and eluted with petroleum ether to give a yellow solid **3s** (100 mg, 87%). Mp: 96.5–98.0 °C. ^1^H NMR (400 MHz, CDCl_3_) *δ* = 8.22 (d, *J* = 7.4 Hz, 2H), 7.53–7.40 (m, 5H), 7.32–7.27 (m, 3H), 7.20 (m, 3H). ^13^C NMR (100 MHz, CDCl_3_) *δ* = 158.8, 152.6, 133.7, 131.1, 129.8, 129.6, 129.4, 129.3, 129.3, 128.7, 127.9, 126.6, 125.6, 120.9, 112.4, 99.3. Analytical data for compound **3s** were consistent with the literature [32].

*2-(4-Bromophenyl)-5-methyl-3-(phenylselanyl)benzofuran* (**3t**): Prepared following general procedure A using substrate 2-((4-bromophenyl)ethynyl)-1-methoxy-4-methylbenzene **1t** (0.3 mmol, 90 mg) and diphenyl diselenide **2a** (0.36 mmol, 112 mg). Isolation was purified by flash chromatography and eluted with petroleum ether to give a white solid **3t** (123 mg, 93%). Mp: 138.6–139.2 °C. ^1^H NMR (400 MHz, CDCl_3_) *δ* = 8.11 (d, *J* = 8.1 Hz, 2H), 7.57 (d, *J* = 8.5 Hz, 2H), 7.44 (d, *J* = 8.3 Hz, 1H), 7.34 (s, 1H), 7.28 (m, 2H), 7.18 (m, 4H), 2.43 (s, 3H). ^13^C NMR (100 MHz, CDCl_3_) *δ* = 156.4, 152.6, 133.4, 132.1, 131.8, 131.4, 129.5, 129.3, 129.2, 129.0, 127.0, 126.4, 123.6, 121.1, 110.9, 100.0, 21.5. HR-MS (ESI) *m*/*z* calcd for C_21_H_16_BrOSe [M + H]^+^ 442.9544, found 442.9534.

*2-Phenyl-3-(phenylselanyl)furo [2,3-b]pyridine (***3u***):* The general procedure **A** was followed using substrate 2-methoxy-3-(phenylethynyl)pyridine **1u** (0.3 mmol, 63 mg) and diphenyl diselenide **2a** (0.36 mmol, 112 mg). Isolation was purified by flash chromatography and eluted with petroleum ether to give a brown oil **3u** (68 mg, 65%). ^1^H NMR (400 MHz, CDCl_3_) *δ* = 8.34 (d, *J* = 4.9 Hz, 1H), 8.30–8.24 (m, 2H), 7.77 (d, *J* = 7.7 Hz, 1H), 7.53–7.41 (m, 3H), 7.30 (m, 3H), 7.25–7.13 (m, 4H). ^13^C NMR (100 MHz, CDCl_3_) *δ* = 161.3, 156.9, 145.0, 130.7, 130.2, 130.0, 129.7, 129.6, 129.5, 128.7, 128.1, 126.8, 124.4, 120.1, 99.0. HR-MS (ESI) *m*/*z* calcd for C_19_H_13_NNaOSe [M + Na]^+^ 374.0055, found 374.0069.

*2,3-bis(Phenylselanyl)benzofuran* (**3v**): Prepared following general procedure **A** using substrate ((2-methoxyphenyl)ethynyl)trimethylsilane **1v** (0.3 mmol, 61 mg) and diphenyl diselenide **2a** (0.36 mmol, 112 mg). Isolation was purified by flash chromatography and eluted with petroleum ether to give a yellow oil **3v** (98 mg, 76%). ^1^H NMR (400 MHz, CDCl_3_) *δ* = 7.61–7.55 (m, 2H), 7.50 (dd, *J* = 16.4, 8.0 Hz, 2H), 7.37 (m, 3H), 7.30 (m, 4H), 7.25–7.17 (m, 3H). ^13^C NMR (100 MHz, CDCl_3_) *δ* = 157.3, 150.9, 132.9, 130.8, 130.6, 130.5, 129.5, 129.3, 129.1, 128.0, 126.8, 125.5, 123.6, 121.1, 113.8, 111.5. Analytical data for compound **3v** were consistent with the literature [56].

*2-(2-Methoxyphenyl)-3-(phenylselanyl)benzofuran* (**3w**): Prepared following general procedure **A** using substrate 2-((2-methoxyphenyl)ethynyl)phenol **1w** (0.3 mmol, 67 mg) and diphenyl diselenide **2a** (0.36 mmol, 112 mg). Isolation was purified by flash chromatography and eluted with petroleum ether to give a yellow solid **3w** (100 mg, 88%). ^1^H NMR (400 MHz, CDCl_3_) *δ* = 7.60 (m, *J* = 8.5, 4.1 Hz, 2H), 7.46 (m, *J* = 13.4, 8.0 Hz, 2H), 7.35 (m, *J* = 9.4, 7.0 Hz, 3H), 7.25–7.15 (m, 4H), 7.12–7.01 (m, 2H), 3.80 (s, 3H). ^13^C NMR (100 MHz, CDCl_3_) *δ* = 157.8, 156.6, 155.0, 131.9, 131.8, 131.4, 130.9, 129.7, 129.2, 129.1, 126.1, 124.8, 123.2, 121.1, 120.4, 119.3, 111.4, 103.2, 55.5. HR-MS (ESI) *m*/*z* calcd for C_21_H_17_O_2_Se [M + H]^+^ 381.0383, found 381.0385

### 4.4. Procedure for the Electrosynthesis of Compounds ***4***

In an undivided cell (10 mL) equipped with a stirring bar, a mixture of substrates **1a** (0.3 mmol), **2** (0.36 mmol), *n*-Bu_4_NPF_6_ (0.5 mmol) and MeCN/HFIP = 4:1 (5 mL) were added. The cell was equipped with a graphite plate (1.5 cm × 1.0 cm × 0.2 cm) as the anode and platinum plate (1.0 cm × 1.0 cm × 0.01 cm) as the cathode connected to an AXIOMET AX-3003P DC regulated power supply. The reaction mixture was stirred and electrolyzed at a constant current of 10 mA at room temperature for 3 h. Upon completion, the solvent was removed directly under reduced pressure to afford the crude product, which was further purified by flash column chromatography to afford the desired products **4a**–**4****m**.

*2-Phenyl-3-(p-tolylselanyl)benzofuran* (**4a**): Prepared following general procedure **B** using substrate 1-methoxy-2-(phenylethynyl)benzene **1a** (0.3 mmol, 62 mg) and 1,2-di-*p*-tolyldiselenide **2a** (0.36 mmol, 122 mg). Isolation was purified by flash chromatography and eluted with petroleum ether to give a yellow solid **4a** (93 mg, 85%). Mp: 84.5–87.7 °C. ^1^H NMR (400 MHz, CDCl_3_) *δ* = 8.27–8.17 (m, 2H), 7.54 (m, 2H), 7.47 (m, 2H), 7.44–7.37 (m, 1H), 7.37–7.30 (m, 1H), 7.25–7.18 (m, 3H), 6.99 (d, *J* = 7.9 Hz, 2H), 2.26 (s, 3H). ^13^C NMR (100 MHz, CDCl_3_) *δ* = 157.1, 154.2, 136.4, 132.1, 130.3, 130.3, 129.7, 129.4, 128.6, 127.9, 127.6, 125.3, 123.5, 121.4, 111.3, 100.2, 21.1. Analytical data for compound **4a** were consistent with the literature [55].

*3-((4-Fluorophenyl)selanyl)-2-phenylbenzofuran* (**4b**): Prepared following general procedure **B** using substrate 1-methoxy-2-(phenylethynyl)benzene **1a** (0.3 mmol, 62 mg) and 1,2-bis(4-fluorophenyl)diselenide **2b** (0.36 mmol, 125 mg). Isolation was purified by flash chromatography and eluted with petroleum ether to give a yellow oil **4b** (91 mg, 83%). ^1^H NMR (400 MHz, CDCl_3_) *δ* = 8.26–8.10 (m, 2H), 7.55 (d, *J* = 8.2 Hz, 1H), 7.53–7.43 (m, 3H), 7.40 (m, 1H), 7.36–7.26 (m, 3H), 7.23 (m, 1H), 6.87 (m, 2H). ^13^C NMR (100 MHz, CDCl_3_) *δ* = 162.0 (d, ^1^*J*_C-F_ = 244.0 Hz), 155.7 (d, ^2^*J*_C-F_ = 291.0 Hz), 131.8, 131.5 (d, ^3^*J*_C-F_ = 8.0 Hz), 130.2, 129.5, 128.6, 127.9, 127.8, 125.7 (d, ^4^*J*_C-F_ = 3.0 Hz), 125.4, 123.6, 121.2, 116.6 (d, ^2^*J*_C-F_ = 22.0 Hz), 111.4, 100.2. ^19^F NMR (376 MHz, CDCl_3_) *δ* = −115.7. Analytical data for compound **4b** were consistent with the literature [32].

*3-((4-Chlorophenyl)selanyl)-2-phenylbenzofuran* (**4c**): Prepared following general procedure **B** using substrate 1-methoxy-2-(phenylethynyl)benzene **1a** (0.3 mmol, 62 mg) and 1,2-bis(4-chlorophenyl)diselenide **2c** (0.36 mmol, 137 mg). Isolation was purified by flash chromatography and eluted with petroleum ether to give a yellow solid **4c** (105 mg, 91%). Mp: 86.8–87.4 °C. ^1^H NMR (400 MHz, CDCl_3_) *δ* = 8.22 (d, *J* = 7.6 Hz, 2H), 7.60 (d, *J* = 8.3 Hz, 1H), 7.55–7.41 (m, 4H), 7.38 (m, 1H), 7.31–7.27 (m, 1H), 7.26–7.22 (m, 2H), 7.18–7.13 (m, 2H). ^13^C NMR (100 MHz, CDCl_3_) *δ* = 157.5, 154.3, 132.5, 131.7, 130.6, 130.1, 129.7, 129.6, 129.6, 128.7, 127.9, 125.5, 123.7, 121.2, 111.4, 99.5. Analytical data for compound **4c** were consistent with the literature [55].

*3-((4-Bromophenyl)selanyl)-2-phenylbenzofuran* (**4d**): Prepared following general procedure **B** using substrate 1-methoxy-2-(phenylethynyl)benzene **1a** (0.3 mmol, 62 mg) and 1,2-bis(4-bromophenyl)diselenide **2d** (0.36 mmol, 169 mg). Isolation was purified by flash chromatography and eluted with petroleum ether to give a yellow solid **4d** (107 mg, 83%). Mp: 98.1–99.7 °C. ^1^H NMR (400 MHz, CDCl_3_) *δ* = 8.18 (d, *J* = 7.5 Hz, 2H), 7.57 (d, *J* = 8.2 Hz, 1H), 7.46 (m, 5H), 7.35 (m, 1H), 7.27 (m, 1H), 7.24 (m, 1H), 7.14 (m, 2H). ^13^C NMR (100 MHz, CDCl_3_) *δ* = 157.6, 154.3, 132.4, 131.7, 130.8, 130.5, 130.0, 129.6, 128.7, 127.9, 125.5, 123.7, 121.1, 120.4, 111.4, 99.4. Analytical data for compound **4d** were consistent with the literature [32].

*2-Phenyl-3-(m-tolylselanyl)benzofuran* (**4e**): Prepared following general procedure **B** using substrate 1-methoxy-2-(phenylethynyl)benzene **1a** (0.3 mmol, 62 mg) and 1,2-di-*m*-tolyldiselenide **2e** (0.36 mmol, 122 mg). Isolation was purified by flash chromatography and eluted with petroleum ether to give a yellow solid **4e** (95 mg, 87%). Mp: 66.5–67.1 °C. ^1^H NMR (400 MHz, CDCl_3_) *δ* = 8.27 (m, 2H), 7.59 (m, 2H), 7.49 (m, 2H), 7.46–7.41 (m, 1H), 7.40–7.35 (m, 1H), 7.28 (m, 1H), 7.20 (s, 1H), 7.14–7.06 (m, 2H), 6.99 (d, *J* = 7.1 Hz, 1H), 2.27 (s, 3H). ^13^C NMR (100 MHz, CDCl_3_) *δ* = 157.3, 154.2, 139.2, 132.1, 131.3, 130.3, 129.8, 129.4, 129.3, 128.6, 127.9, 127.3, 126.3, 125.3, 123.5, 121.4, 111.3, 99.9, 21.5. Analytical data for compound **4e** were consistent with the literature [32].

*3-((3-Chlorophenyl)selanyl)-2-phenylbenzofuran* (**4f**): Prepared following general procedure **B** using substrate 1-methoxy-2-(phenylethynyl)benzene **1a** (0.3 mmol, 62 mg) and 1,2-bis(3-chlorophenyl)diselenide **2f** (0.36 mmol, 137 mg). Isolation was purified by flash chromatography and eluted with petroleum ether to give a yellow solid **4f** (100 mg, 87%). Mp: 58.4–59.2 °C. ^1^H NMR (400 MHz, CDCl_3_) *δ* = 8.20 (d, *J* = 7.5 Hz, 2H), 7.59 (d, *J* = 8.2 Hz, 1H), 7.48 (m, 4H), 7.38 (m, 1H), 7.32–7.26 (m, 2H), 7.14 (m, 2H), 7.10–7.05 (m, 1H). ^13^C NMR (100 MHz, CDCl_3_) *δ* = 157.7, 154.3, 135.2, 133.4, 131.7, 130.4, 130.0, 129.6, 128.7, 128.7, 127.9, 127.1, 126.6, 125.5, 123.7, 121.1, 111.4, 99.1. HR-MS (ESI) *m*/*z* calcd for C_20_H_14_ClOSe [M + H]^+^ 384. 9893, found 384. 9887.

*3-((3-Bromophenyl)selanyl)-2-phenylbenzofuran* (**4g**): Prepared following general procedure **B** using substrate 1-methoxy-2-(phenylethynyl)benzene **1a** (0.3 mmol, 62 mg) and 1,2-bis(3-bromophenyl)diselenide **2g** (0.36 mmol, 169 mg). Isolation was purified by flash chromatography and eluted with petroleum ether to give a yellow solid **4g** (103 mg, 80%). Mp: 56.0–57.5 °C. ^1^H NMR (400 MHz, CDCl_3_) *δ* = 8.20 (d, *J* = 7.3 Hz, 2H), 7.59 (d, *J* = 8.2 Hz, 1H), 7.54–7.45 (m, 4H), 7.45–7.42 (m, 1H), 7.40–7.34 (m, 1H), 7.31–7.26 (m, 2H), 7.17 (d, *J* = 7.9 Hz, 1H), 7.01 (m, 1H). ^13^C NMR (100 MHz, CDCl_3_) *δ* = 157.7, 154.3, 133.7, 131.7, 131.5, 130.8, 130.0, 129.6, 129.5, 128.7, 127.9, 127.6, 125.5, 123.7, 123.4, 121.1, 111.4, 99.1. HR-MS (ESI) *m*/*z* calcd for C_20_H_14_BrOSe [M + H]^+^ 428.9388, found 428.9490.

*2-Phenyl-3-((3-(trifluoromethyl)phenyl)selanyl)benzofuran* (**4h**): Prepared following general procedure **B** using substrate 1-methoxy-2-(phenylethynyl)benzene **1a** (0.3 mmol, 62 mg) and 1,2-bis(3-(trifluoromethyl)phenyl)diselenide **2h** (0.36 mmol, 161 mg). Isolation was purified by flash chromatography and eluted with petroleum ether to give a yellow solid **4h** (80 mg, 64%). Mp: 84.6–85.1 °C. ^1^H NMR (400 MHz, CDCl_3_) *δ* = 8.18 (d, *J* = 7.5 Hz, 2H), 7.64–7.56 (m, 2H), 7.46 (m, 4H), 7.42–7.33 (m, 3H), 7.28 (m, 1H), 7.23 (m, 1H). ^13^C NMR (100 MHz, CDCl_3_) *δ* = 157.9, 154.3, 132.9, 132.2, 131.8, 131.5 (d, ^3^*J*_C-F_ = 8.0 Hz), 130.0, 129.8 (d, ^3^*J*_C-F_ = 8.0 Hz), 128.7, 127.9, 125.7 (d, ^3^*J*_C-F_ = 6.0 Hz), 125.6, 125.1, 123.8, 123.2 (q, ^3^*J*_C-F_ = 7.0 Hz), 122.4, 121.1, 111.5, 98.9. ^19^F NMR (376 MHz, CDCl_3_) *δ* = −62.7. Analytical data for compound **4h** were consistent with the literature [57].

*2-Phenyl-3-(o-tolylselanyl)benzofuran* (**4i**): Prepared following general procedure **B** using substrate 1-methoxy-2-(phenylethynyl)benzene **1a** (0.3 mmol, 62 mg) and 1,2-di-*o*-tolyldiselenide **2i** (0.36 mmol, 122 mg). Isolation was purified by flash chromatography and eluted with petroleum ether to give a colorless liquid **4i** (98 mg, 90%). ^1^H NMR (400 MHz, CDCl_3_) *δ* = 8.23 (d, *J* = 7.4 Hz, 2H), 7.62 (d, *J* = 8.3 Hz, 1H), 7.48 (m, 4H), 7.39 (m, 1H), 7.29 (d, *J* = 7.3 Hz, 1H), 7.22 (d, *J* = 7.4 Hz, 1H), 7.11 (m, 1H), 7.02 (d, *J* = 7.7 Hz, 1H), 6.93 (m, 1H), 2.53 (s, 3H). ^13^C NMR (100 MHz, CDCl_3_) *δ* = 157.7, 154.4, 136.8, 132.2, 132.1, 130.3, 130.2, 129.4, 128.6, 128.4, 127.9, 126.9, 126.2, 125.4, 123.6, 121.4, 111.3, 99.2, 21.6. Analytical data for compound **4i** were consistent with the literature [55].

*3-((2-Chlorophenyl)selanyl)-2-phenylbenzofuran* (**4j**): Prepared following general procedure **B** using substrate 1-methoxy-2-(phenylethynyl)benzene **1a** (0.3 mmol, 62 mg) and 1,2-bis(2-chlorophenyl)diselenide **2j** (0.36 mmol, 137 mg). Isolation was purified by flash chromatography and eluted with petroleum ether to give a yellow oil **4j** (108 mg, 94%). ^1^H NMR (400 MHz, CDCl_3_) *δ* = 8.18 (d, *J* = 7.3 Hz, 2H), 7.61 (d, *J* = 8.2 Hz, 1H), 7.54 (d, *J* = 7.9 Hz, 1H), 7.49–7.35 (m, 5H), 7.29 (m, 1H), 7.13–7.05 (m, 1H), 6.94 (m, 1H), 6.85 (m, 1H). ^13^C NMR (100 MHz, CDCl_3_) *δ* = 158.4, 154.4, 132.7, 132.2, 131.8, 130.0, 129.7, 129.6, 128.8, 128.7, 127.9, 127.7, 127.0, 125.6, 123.8, 121.3, 111.4, 98.6. Analytical data for compound **4j** were consistent with the literature [32].

*2-Phenyl-3-(thiophen-2-ylselanyl)benzofuran* (**4k**): Prepared following general procedure **B** using substrate 1-methoxy-2-(phenylethynyl)benzene **1a** (0.3 mmol, 62 mg) and dimethyldiselenide **2k** (0.36 mmol, 117 mg). Isolation was purified by flash chromatography and eluted with petroleum ether to give a yellow solid **4k** (83 mg, 83%). Mp: 90.4–92.1 °C. ^1^H NMR (400 MHz, CDCl_3_) *δ* = 8.29 (d, *J* = 7.6 Hz, 2H), 7.72 (d, *J* = 7.5 Hz, 1H), 7.55 (m, 3H), 7.47 (m, 1H), 7.34 (m, 4H), 6.95–6.87 (m, 1H). ^13^C NMR (100 MHz, CDCl_3_) *δ* = 156.3, 154.0, 133.7, 131.8, 131.5, 130.3, 130.0, 129.4, 128.6, 128.1, 127.9, 125.3, 123.5, 121.1, 111.3, 102.2. HR-MS (ESI) m/z calcd for C_18_H_13_OSSe [M + H]^+^ 356.9847, found 356. 9843.

*3-(Methylselanyl)-2-phenylbenzofuran* (**4l**): Prepared following general procedure **B** using substrate 1-methoxy-2-(phenylethynyl)benzene **1a** (0.3 mmol, 62 mg) and dimethyldiselenide **2l** (0.36 mmol, 68 mg). Isolation was purified by flash chromatography and eluted with petroleum ether to give a yellow solid **4l** (64 mg, 74%). Mp: 90.4–92.1 °C. ^1^H NMR (400 MHz, CDCl_3_) *δ* = (m, 2H), 7.70 (m, 1H), 7.57–7.47 (m, 3H), 7.41 (m, 1H), 7.37–7.29 (m, 2H), 2.23 (s, 3H). ^13^C NMR (100 MHz, CDCl_3_) *δ* = 155.5, 154.0, 132.3, 130.7, 129.1, 128.6, 127.8, 125.1, 123.3, 121.0, 111.3, 101.3, 8.4. HR-MS (ESI) *m*/*z* calcd for C_15_H_13_OSe [M + H]^+^ 289.0126, found 289.0122.

*3-(Ethylselanyl)-2-phenylbenzofuran* (**4m**): Prepared following general procedure **B** using substrate 1-methoxy-2-(phenylethynyl)benzene **1a** (0.3 mmol, 62 mg) and diethyldiselenide **2m** (0.36 mmol, 78 mg). Isolation was purified by flash chromatography and eluted with petroleum ether to give a colorless liquid **4m** (57 mg, 63%). ^1^H NMR (400 MHz, CDCl_3_) *δ* = 8.35–8.29 (m, 2H), 7.74–7.67 (m, 1H), 7.56–7.46 (m, 3H), 7.40 (m, 1H), 7.37–7.28 (m, 2H), 2.82 (q, *J* = 7.4 Hz, 2H), 1.33 (t, *J* = 7.4 Hz, 3H). ^13^C NMR (100 MHz, CDCl_3_) *δ* = 156.2, 154.0, 132.9, 130.8, 129.0, 128.5, 127.8, 125.1, 123.3, 121.2, 111.2, 100.1, 22.1, 16.0. HR-MS (ESI) *m*/*z* calcd for C_16_H_15_OSe [M + H]^+^ 303.0283, found 303.0279.

### 4.5. Procedure for the Scaled-Up Synthesis of Compound ***3a*** and ***3t***

In an undivided cell (250 mL) equipped with a stirring bar, a mixture of substrates **1a** (15.0 mmol, 3.12 g), **2a** (18.0 mmol, 5.62 g), *n*-Bu_4_NPF_6_ (25.0 mmol, 9.69 g) and MeCN/HFIP = 4:1 (150 mL) were added. The cell was equipped with a graphite plate (3 cm × 3 cm × 0.6 cm) as the anode and platinum plate (3 cm × 3 cm × 0.01 cm) as the cathode and connected to a DC regulated power supply. The reaction mixture was stirred and electrolyzed at a constant current of 50 mA at 23 °C bath for 14 h. When the reaction was finished, the mixture was concentrated under reduced pressure. Purification by column chromatography on silica gel (eluent: petroleum ether) yielded solo product **3a** (4.45 g, 85%) as yellow solid.

In an undivided cell (250 mL) equipped with a stirring bar, a mixture of substrates **1****t** (5.0 mmol, 1.50 g), **2a** (6.0 mmol, 1.87 g), *n*-Bu_4_NPF_6_ (8.3 mmol, 3.23 g) and MeCN/HFIP = 4:1 (150 mL) were added. The cell was equipped with a graphite plate (3 cm × 3 cm × 0.6 cm) as the anode and platinum plate (3 cm × 3 cm × 0.01 cm) as the cathode and connected to a DC regulated power supply. The reaction mixture was stirred and electrolyzed at a constant current of 50 mA at 23 °C bath for 5 h. When the reaction was finished, the mixture was concentrated under reduced pressure. Purification by column chromatography on silica gel (eluent: petroleum ether) yielded solo product **3****t** (1.86 g, 84%) as white solid.

## Data Availability

Not applicable.

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
