# Peer review of "Direct Electrooxidative Selenylation/Cyclization of Alkynes: Access to Functionalized Benzo[b]furans"

_molecules, 2022, doi:10.3390/molecules27196314_

Round 1
Reviewer 1 Report
The manuscript entitled “Direct Electrooxidative Selenylation/Cyclization of Alkynes: Access to Functionalized Benzo[b]furans” describes the titular study and provides a range of substrate scopes for the cyclization to a benzofuran system with descent to high yields. This is an advancement of the authors’ earlier observations (J. Org. Chem. 2021, 86, 16044). I recommend this manuscript for publication after major revisions.
Comments:
1. Authors are suggested to explain the following in the revised manuscript:
(a) What is the source of the hydroxyl anion in the mechanism?
(b) Why authors have used HFIP alongside acetonitrile?
2. NMR: The authors are suggested to provide 77Se NMR for at least few of the unknown molecules
3. Citations: The authors are suggested to cite these recent works on diselenides
(a) Zade et al. Chem. Commun., 2014, DOI: 10.1039/C4CC05439C
(b) Zade et al. ACS Applied Materials & Interfaces 2016, 8, 18222.
(c) H.-Y. Liu, J.-R. Zhang, G.-B. Huang, Y.-H. Zhou, Y.-Y. Chen, Y.-L. Xu, Adv. Synth. Catal. 2021, 363, 1656.
Author Response
Dear reviewer and editor,
Thank you for your precious and supportive comments to our manuscript. We have revised the manuscript and updated the SI according to your suggestions, and the changes we have made also marked using the “Track Changes”function.
Please see the attachment.

Reviewer 2 Report
The manuscript by Ruan et al. demonstrates a direct electrooxidative Selenylation/Cyclization of Alkenes. Authors developed a new method and synthesized a few interesting products in 45-96% yield. The manuscript is worth publishing after minor revision.
1/ Please explain, why 3k has so low yield if 3l has 94%. Did you observe a side reaction?
2/ Compounds 3l and 3m are missing HRMS data.
3/ Please explain a reaction 5d. Compound 3x has not been observed. Did you recover substrate or observe a side reaction?
Author Response
Dear referees and editor,
Thank you for your precious and supportive comments to our manuscript. We have revised the manuscript and updated the SI according to your suggestions, and the changes we have made also marked by using the “Track Changes” Function under the MS Word.
Here is the response point by point.
Please see the attachment.

Reviewer 3 Report
The Ruan research group looks quite active in the area of Electroorganic synthesis as evident from the previous research publications of the group in this area. The group has presented a nice method of synthesizing Functionalized Benzo[b]furans through Electrooxidative Selenylation/Cyclization of Alkynes and, the results presented here looks quite promising. The results of the present study will be beneficial for the researchers working in the area of organic synthesis, in particular metal-free oxidative transformations.
I would recommend the publication of this manuscript, however, the manuscript needs to be edited as far as English language is concerned. Moreover, in the control experiments for the mechanisitic pathway, the authors reported the formation of sole product 3w (88%) in the case of phenols. Here, what about the rest of 12%, moreover, if this is a sole product, how the authors ruled out the possibility of Pathway II for this experiment?
Author Response

(The authors gave the same response as above.)

Reviewer 4 Report
The manuscript “Direct Electrooxidative Selenylation/Cyclization of Alkynes: Access to Functionalized Benzo[b]furans" describes the synthesis of differently functionalized 3-organoselenyl-benzo[b]furans. The works is robust and the authors have efficiently synthesized 35 examples obtained in moderate to excellent yields via electrooxidative selenylation/cyclization of 2-alkynylanisoles. Plausible mechanisms were proposed for the formation of the products and experiments were carried out to clarify the reaction pathway. All products were well characterized by NMR and the spectra have been presented.
Thus, this manuscript should be accepted in Molecules after revision of some aspects:
Thus, this manuscript should be accepted in Molecules after revision of some aspects:
The chemistry of this work is very similar to that one described by Braga and co-workers (Frontiers in Chemistry, 2022, 10, 880099). Thus, the authors must mentioned this article and clarify the differences between these studies by comparing the methodologies.
Introduction: some references regarding the use transition-metal and diorganyl dichalcogenides to promote cyclization were not mentioned. For example, the Eur. J. Org. Chem., 2017, 6382, which is related to the synthesis of 3-organoselenyl-benzofurans using CuI and diorganyl diselenides, the Synthesis, 2021, 53, 2545, a revision paper about the use copper and iron salts with diorganyl dichalcogenides to obtain carbo- and heterocycles by cyclization of alkynes, and the Chemical Record, 2021, 21, 1342, related to the metal-free synthesis of hetero- and carbocycles via cyclization of alkynes with organylchalcogenides.
Scheme 1: the Ph must be removed from R1 since this example did not work; the X must be defined and added to the legend; I suggest changing the R by R3 in order to create a logical sequence; change the “selenenylbenzofurans” by “selenylbenzofurans”.
Page 3, line 74: the HFIP must be written in full.
Table 1: why different amounts of diphenyl diselenide were not evaluated? The use of less than one equivalent of the diselenide should clarify some important aspects by including mechanistic ones. Since the product 3v was obtained in 76% with the incorporation of both PhSe portions at the final product structure and the control experiments have shown two probable pathways promoted by different selenium species (radical and cationic), only half equivalent of the diorganyl diselenide could be sufficient to complete the transformation. The reduction of the diselenide amount should be very interesting, especially for scaled-up reactions. I recommend setting up an experiment in this sense.
Scheme 6: it is not clear the source of the hydroxyl groups which are responsible for the demethylation process. Could it the HFIP? In the reaction in absence of HFIP or MeOH (table 1, entry 3), how the demethylation occurs?
Author Response

(The authors gave the same response as above.)

Round 2
Reviewer 1 Report
The authors carefully addressed the previously mentioned issues. Therefore, I suggest acceptance of this revised manuscript.
Reviewer 4 Report
Considering the authors have taken in count most of the aspects and suggestions from this reviewer and have also modified the manuscript to improve its quality for the best understanding of the readers I recommend this paper for publication in Molecules without any other changes.